# Seasonally migratory songbirds have different historic population size characteristics than resident relatives

Kevin Winker[1]*, Kira Delmore[2,3]

[1]University of Alaska Museum and Department of Biology and Wildlife, Fairbanks, United States; [2]Department of Biology, Texas A&M University, College Station, United States; [3]Ecology, Evolution, and Environmental Biology, Columbia University, New York, United States

## eLife Assessment

This is a **valuable** study of the role that life history differences might play in determining population size and demography. While concerns about generation times and population structure leave the evidence for the claims in parts **incomplete**, the work is of considerable interest to anyone who tries to understand evolutionary consequences of life history changes.

*For correspondence:
kevin.winker@alaska.edu

**Abstract** Modern genomic methods enable estimation of a lineage's long-term effective population sizes back to its origins. This ability allows unprecedented opportunities to determine how the adoption of a major life-history trait affects lineages' populations relative to those without the trait. We used this novel approach to study the population effects of the life-history trait of seasonal migration across evolutionary time. Seasonal migration is a common life-history strategy, but its effects on long-term population sizes relative to lineages that don't migrate are largely unknown. Using whole-genome data, we estimated effective population sizes over millions of years in closely related seasonally migratory and resident lineages in a group of songbirds. Our main predictions were borne out: Seasonal migration is associated with larger effective population sizes ($N_e$), greater long-term variation in $N_e$, and a greater degree of initial population growth than among resident lineages. Initial growth periods were remarkably long (0.63–4.29 Myr), paralleling the expansion and adaptation phases of taxon cycles, a framework of lineage expansion and eventual contraction over time encompassing biogeography and evolutionary ecology. Heterogeneity among lineages is noteworthy, despite geographic proximity (including overlap) and close relatedness. Seasonal migration imbues these lineages with fundamentally different population size attributes through evolutionary time compared to closely related resident lineages.

## Introduction

Modern genomic methods enable estimation of a lineage's long-term effective population sizes back to its origins (*Li and Durbin, 2011*; *Mazet et al., 2015*; *Nadachowska-Brzyska et al., 2015*). This ability allows unprecedented opportunities to study the populational effects of past evolutionary and climatic phenomena on organismal lineages, and this new area of paleodemographics is likely to grow rapidly (*Nadachowska-Brzyska et al., 2022*, *Germain et al., 2023a*; *Germain et al., 2023b*). We use this approach in a novel way to test hypotheses about how the adoption of a major life-history trait affected lineages' populations relative to those without the trait.

Seasonal migration is a common life-history strategy among the world's birds, but its effects on long-term population sizes relative to lineages that don't have this trait are largely unknown (*Newton, 2007*; *Rappole, 2013*). Demographic differences between seasonally migratory and resident lineages have been studied for decades, but, thus far, ecological, behavioral, and genetic approaches have necessarily been focused on recent and microevolutionary phenomena (e.g. *Greenberg, 1980*; *Tris et al., 2004*; *Sandercock and Jaramillo, 2002*). Here, we look much deeper, contrasting changes in effective population sizes through millions of years between closely related migratory and resident lineages.

Thrushes in the genus *Catharus* and this group's sister *Hylocichla mustelina* (Aves: Turdidae) are a model songbird system for studying the evolutionary effects of seasonal migration. This group has both migratory and resident lineages (*Figure 1*), and it has been important in the study of seasonal migration, divergence, and speciation (e.g. *Blain et al., 2024*; *Delmore et al., 2015*, *Delmore et al., 2016*; *Everson et al., 2019*; *Justen et al., 2024*; *Outlaw et al., 2003*; *Ruegg et al., 2014*; *Termignoni-Garcia et al., 2022*; *Voelker et al., 2013*; *Winker, 2000*; *Winker, 2010*; *Winker and Pruett, 2006*). These animals are relatively small, omnivorous or insectivorous, forest-related birds with lineages that are long-distance seasonal migrants in North America and others that are resident in tropical North and South America (*Collar, 2005*). The current breeding grounds of the migratory lineages span the Russian Far East, Alaska, northern Canada, and the United States, and their wintering ranges include Mexico, the Caribbean, Central America, and tropical South America. In contrast, the resident lineages occupy Mexico, Central America, and northwestern South America (*Collar, 2005*; *Figure 2*).

We made three predictions about effective population sizes ($N_e$) in seasonally migratory vs. resident lineages of these thrushes. First, we can think of North America roughly as an inverted triangle, with substantially more geographic space in the north than in the south. Among these thrushes, the migratory lineages breed in that northern, on average, larger space. The resident lineages instead currently occupy, on average, smaller geographic ranges, even in South America, where they are largely montane (*Figure 2*). Thus, larger breeding ranges among migrants, on average, should yield higher population sizes ($N_e$).

Second, climatic variability is also higher with increased latitude, often temporarily rendering vast swaths of northern North America unoccupiable to many long-distance migrant species (e.g. *Colinvaux et al., 2000*; *Pielou, 1991*). Thus, higher-latitude breeding occupancy should cause larger fluctuations in population size, meaning that variation in $N_e$ will on average be higher among seasonal migrants. Finally, seasonal migration is likely to cause enhanced ecological (and in this case also geographic) release (i.e. in the occupancy of new niche space) in relation to resident relatives. Thus, as migratory lineages' mobility enabled them to take advantage of seasonal resource blooms in ecosystems at higher latitudes and expand their breeding ranges (geographic and ecological expansion), their populations should, on average, have grown proportionally more and at a faster rate than sedentary lineages, thus achieving the first prediction.

Here, we test these predictions by reconstructing effective population sizes of these thrush lineages through time using whole-genome data and the pairwise sequentially Markovian coalescent (PSMC; *Li and Durbin, 2011*).

## Results

The focal variables of our hypotheses all lacked phylogenetic signal. Mean effective population sizes and their variation were both higher among migratory lineages, as predicted (*Table 1*; $U=7$ and 10, $p<0.05$ for both; one-tailed Mann-Whitney $U$-test). The degree of early population growth was also proportionally higher among migrants, as predicted (*Table 1*; $U=9$, $p<0.05$; one-tailed Mann-Whitney $U$-test), but the rate of this early growth was not different between the two groups (*Table 1*; $U=21.5$, $p>0.05$).

To examine why the degree of initial migrant population growth was higher but the rate of growth was not, we considered that this might be because that growth extended over a longer period of time. To our surprise, we found that these growth periods were remarkably long, ranging from 0.63 to 4.29 Myr and averaging 2.51 Myr (*Table 1*, *Figure 3*). Migratory lineages did have a longer initial period of growth (migrants mean deltaT = 3.10 Myr vs. resident mean deltaT = 1.72 Myr; *Table 1*; $U=9$, $p<0.05$).

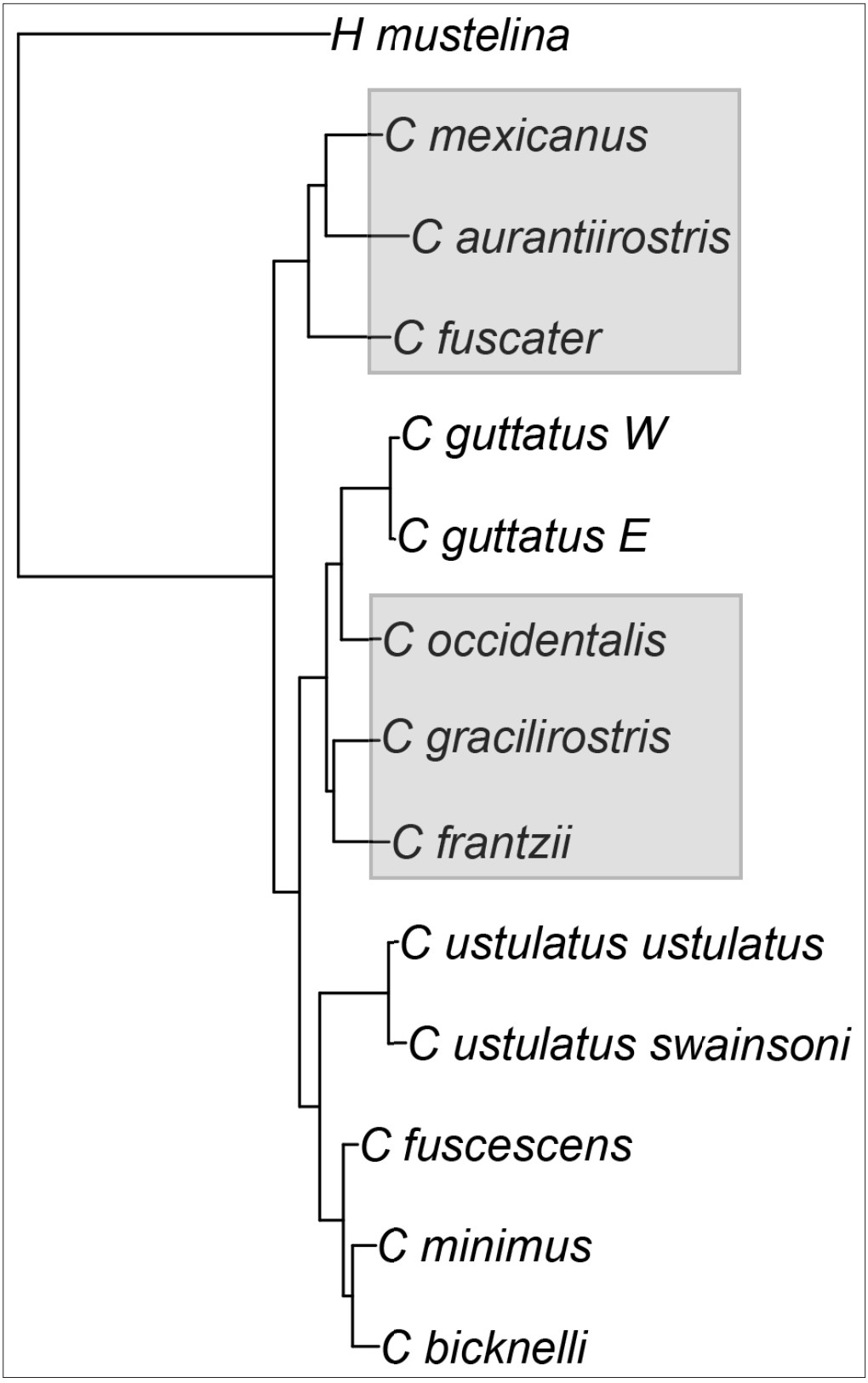

**Figure 1.** The phylogenetic tree of the songbird lineages in this study, from the genera *Catharus* and *Hylocichla*. Neotropical residents are shaded in gray.

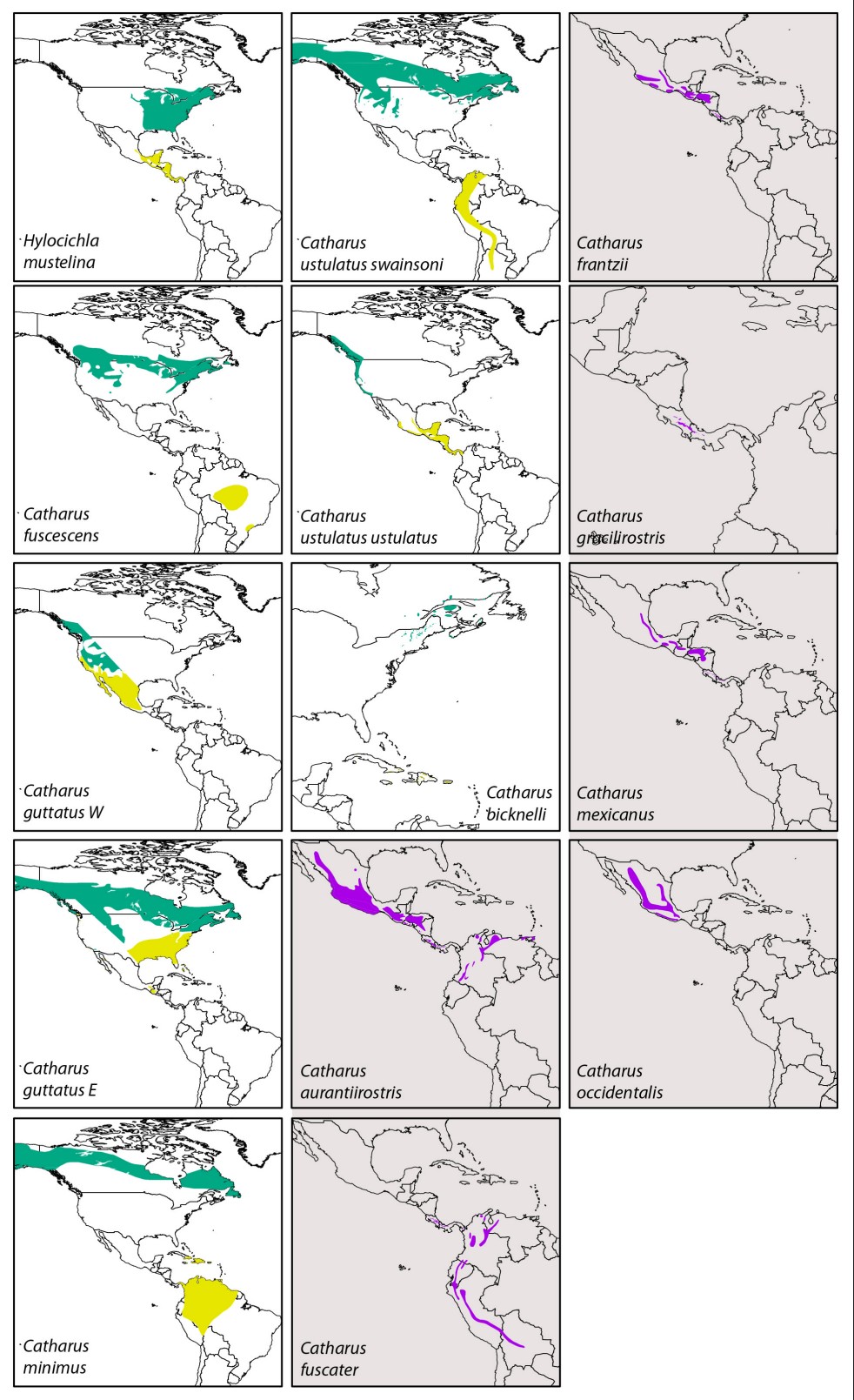

**Figure 2.** Distribution maps of the thrush taxa in this study. Among seasonal migrants, green indicates breeding range and yellow is wintering range. Among sedentary lineages (those shaded in gray), purple indicates year-round range. Data are from **Bird Life International and the Handbook of the Birds of the World, 2021**. Available at http://datazone.birdlife.org/species/requestdis.

**Table 1.** Data from pairwise sequentially Markovian coalescent (PSMC) analyses reflecting effective population sizes ($N_e \times 10^4$) through history at depths > 50 Kyr and the five variables derived and analyzed from that output.
Taxa shaded in gray are Neotropical residents.

| Taxon | Mean $N_e$ (±SD) | SD/mean | Degree of early growth 1 - ($N_{trough}/N_{peak}$) | Rate of early growth degree/ deltaT | deltaT |
|---|---|---|---|---|---|
| *H. mustelina* | 35.63 (±20.11) | 0.56 | 0.86 | 2.92E-07 | 2,946,862 |
| *Catharus fuscescens* | 95.59 (±50.99) | 0.53 | 0.83 | 1.93E-07 | 4,284,020 |
| *Catharus guttatus E* | 75.34 (±65.69) | 0.87 | 0.90 | 4.48E-07 | 2,021,768 |
| *Catharus guttatus W* | 39.72 (±24.20) | 0.61 | 0.81 | 3.28E-07 | 2,477,684 |
| *Catharus minimus* | 63.71 (±28.83) | 0.45 | 0.78 | 2.17E-07 | 3,594,914 |
| *Catharus ustulatus swainsonii* | 76.61 (±66.61) | 0.87 | 0.79 | 2.12E-07 | 3,740,764 |
| *Catharus ustulatus ustulatus* | 39.56 (±12.15) | 0.31 | 0.63 | 2.14E-07 | 2,943,763 |
| *Catharus bicknelli* | 46.90 (±24.12) | 0.51 | 0.78 | 2.79E-07 | 2,781,554 |
| *Catharus aurantiirostris* | 12.38 (±2.83) | 0.23 | –0.75 | –1.00E-06 | 747,451 |
| *Catharus fuscater* | 11.38 (±1.66) | 0.15 | 0.39 | 6.18E-07 | 627,449 |
| *Catharus frantzii* | 24.71 (±8.41) | 0.34 | 0.17 | 2.20E-07 | 774,024 |
| *Catharus gracilirostris* | 29.98 (±9.28) | 0.31 | 0.58 | 3.52E-07 | 1,637,663 |
| *Catharus mexicanus* | 37.10 (±16.18) | 0.44 | 0.75 | 3.34E-07 | 2,250,295 |
| *Catharus occidentalis* | 76.40 (±53.14) | 0.70 | 0.91 | 2.11E-07 | 4,292,405 |
| **Means (±SD)** | | | | | |
| Migrants | 59.34 (±35.70)* | 0.57 (±0.20)* | 0.798 (±0.08)* | 2.73E-7 (±0.85E-7) | 3,098,906 (±732,563)* |
| Residents | 31.99 (±15.25) | 0.36 (±0.19) | 0.341 (±0.59) | 1.23E-7 (±5.69E-7) | 1,721,547 (+1,409,939) |

*p<0.05.

In addition to testing our hypotheses, our results reveal other important attributes of these songbirds. First, they appear to show few of the general population declines at the beginning of the last glacial period (~115 Kya) that *Nadachowska-Brzyska et al., 2016*, found among a global sampling of birds (*Figures 4 and 5*). Second, and more importantly, there is a general lack of temporal synchrony in historic population size changes (*Figures 4 and 5*), despite considerable geographic proximity and overlap of ranges (especially during the nonbreeding season). Finally, a pattern widely shared in the group is that most of the recent populations are below each lineage's historic peak, and this trend predates the arrival of humans in the New World (*Figures 4 and 5*). A striking exception is *C. ustulatus swainsonii*, which has had remarkable long-term growth (*Figure 5*).

## Discussion

To our knowledge, this is the first study to contrast long-term effective population size changes among closely related species to test a priori hypotheses about the population effects of a major life-history trait; here, we focused on seasonal migration. This perspective offers several new insights into migrant demographics and evolutionary ecology. Migratory thrush lineages showed higher effective population sizes, greater population size variation, and proportionally larger initial population growth than resident relatives. A migratory life-history strategy thus imbues these lineages with population size characteristics that are fundamentally different, on average, from those of resident relatives.

Although migratory thrush lineages showed higher proportional initial growth than resident relatives, this was not achieved through rapid adaptation to new geographic and ecological opportunities as we predicted. Population growth in migrants and residents was positive early in the lineages' histories (*deltaT*) in all cases but one (*C. aurantiirostris*), and it occurred over fairly long periods of time

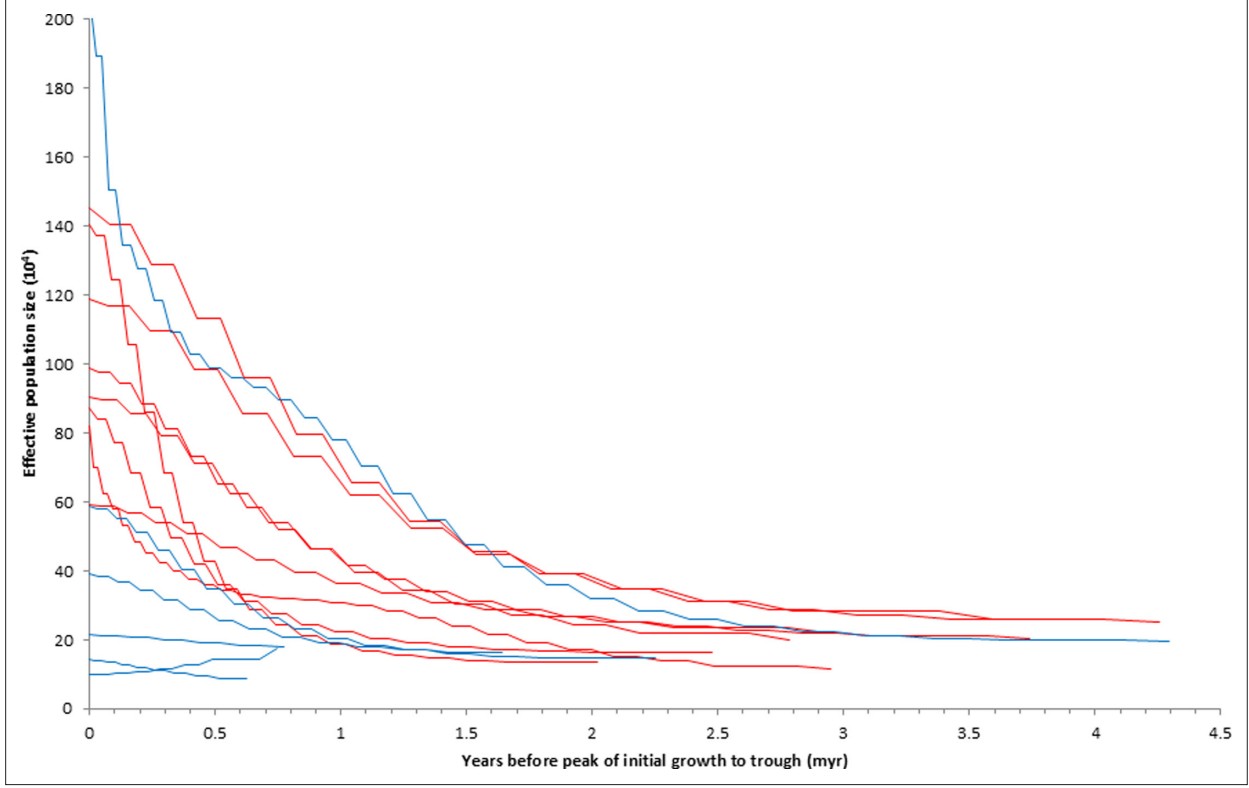

**Figure 3.** Partial historic effective population size curves from all lineages in this study, based on pairwise sequentially Markovian coalescent (PSMC) analyses. Each peak of initial growth is set to zero years to set a common framework in which to visualize the periods and magnitudes of initial growth among migrant (red) and resident (blue) lineages. See specific lineages with their bootstrapped results in *Figure 5*.

(*Table 1*, *Figures 3 and 5*). That this process initially (on average) involves millions of years of population growth in this group, apparently extended by a migratory life-history strategy that provided access to higher latitudes (*Table 1*; *Appendix 1—table 1*), is an insight from our study that should be examined in future work from a broader taxonomic perspective. While actual census population sizes would likely have shown considerable variation across this time, especially after the glacial cycles of the Pleistocene began ~2.6 Mya, the long-term effective population sizes of migrants tended to show steady increases, achieving peaks on average almost 3 Myr after the lineages' PSMC inception (*Figures 1 and 5*; *Table 1*).

With *deltaT* being generally long but highly variable (*Table 1*), one can envision in this group a sort of ur-thrush trajectory in which unique, lineage-specific processes occur as each evolves, influenced by geography, ecological factors, and life-history strategy. These long periods of early growth are concordant with the initial expansion or adaptation phases of taxon cycles, a conceptual framework of lineage expansion and contraction over time through interactions between biogeography and evolutionary ecology (*Pepke et al., 2019*; *Ricklefs and Bermingham, 2002*; *Ricklefs and Cox, 1972*; *Wilson, 1961*).

Another noteworthy pattern widely shared in this group is that most recent populations are below each lineage's historic peak (*Figures 4 and 5*). This, too, is reminiscent of taxon cycles (smaller populations in later stages), and this trend both predates the arrival of humans in the New World (*Figure 5*) and reflects a broader pattern of historic avian declines (*Germain et al., 2023b*). Yet another pattern is a general lack of temporal synchrony in historic population size changes, despite considerable geographic proximity and overlap of ranges (*Figures 4 and 5*). In their review of taxon cycles, *Ricklefs and Bermingham, 2002*, inferred from such a lack of similarity among lineages that each is in a unique population-environment relationship, responding idiosyncratically in a coevolutionary relationship with factors such as parasites, predators, pathogens, and competitors.

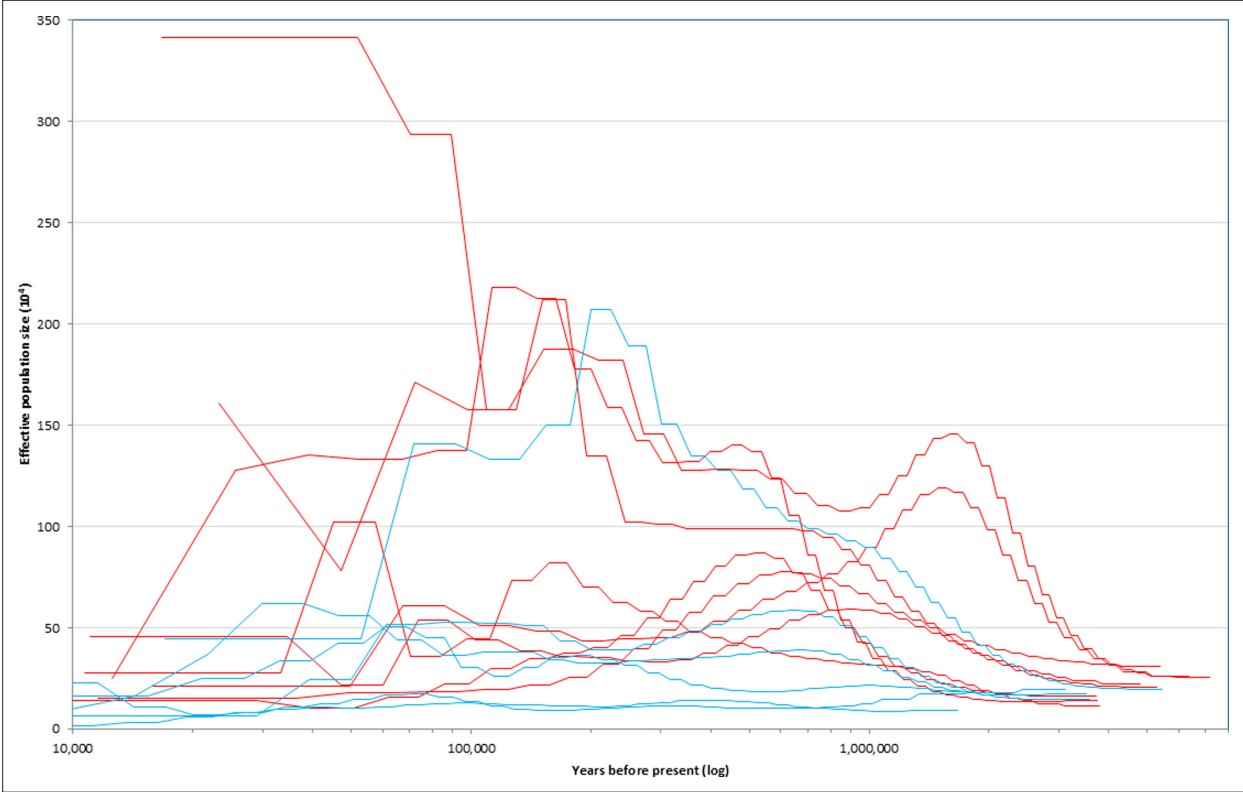

**Figure 4.** Historic effective population size curves from all lineages, based on pairwise sequentially Markovian coalescent (PSMC) analyses. Migrant lineages are in red and resident lineages in blue. See specific lineages with their bootstrapped results in *Figure 5*.

We attribute the higher variation in $N_e$ among migrants to be the result of the relative instability of northern biomes compared with tropical ones through glacial-interglacial cycles (e.g. *Colinvaux et al., 2000*; *Pielou, 1991*).

## Population structure

PSMC estimates of population size can be affected by population structure: a change in structure can produce the signal of a change in size (*Mazet et al., 2015*; *Mazet et al., 2016*; *Wakeley, 1999*). Some of the population size changes we observed might therefore reflect changes only in levels of population connectivity. There are well-understood differences between seasonal migrants and residents in population structure. Migrants generally have lower levels of geographic partitioning of both phenotypic (subspecies) and genetic variation (*Belliure et al., 2000*; *Delmore et al., 2020*; *Mayr, 1963*; *Montgomery, 1896*). The resident lineages we studied are more prone to this structure than the migrant lineages, with migrants having an average of 2.5 subspecies per lineage and residents having an average of 5.9 (*Collar, 2005*). Migratory lineages therefore come closer to meet the assumption of PSMC analyses that populations are panmictic. Thus, in addition to our geographic and ecological scenario differing between the migrant and resident lineages we studied, we also have a pervasive effect of population structure affecting our results, likely increasingly complementing geographic expansion among seasonal migrants as lineage ages approach the recent. However, the greatest effects of inflated $N_e$ over evolutionary time would likely occur among nonmigrants, with more isolation and less mixing (*Li and Durbin, 2011*).

Accounting for population structure in these models to distinguish between the two phenomena is a difficult computational problem; the correlation is not simple, and it might not be tractable with data from a single individual (*Mazet et al., 2016*; *Nadachowska-Brzyska et al., 2016*). However, each individual has a genome amalgamated from the entire lineage metapopulation, including individuals and populations that have not been sampled (*Mazet et al., 2015*). When not focusing on relatively recent or rapid changes (e.g. in human or other bottlenecked systems, where this has been most studied), these issues are probably not as important (*Mazet et al., 2016*). For example, sister lineages

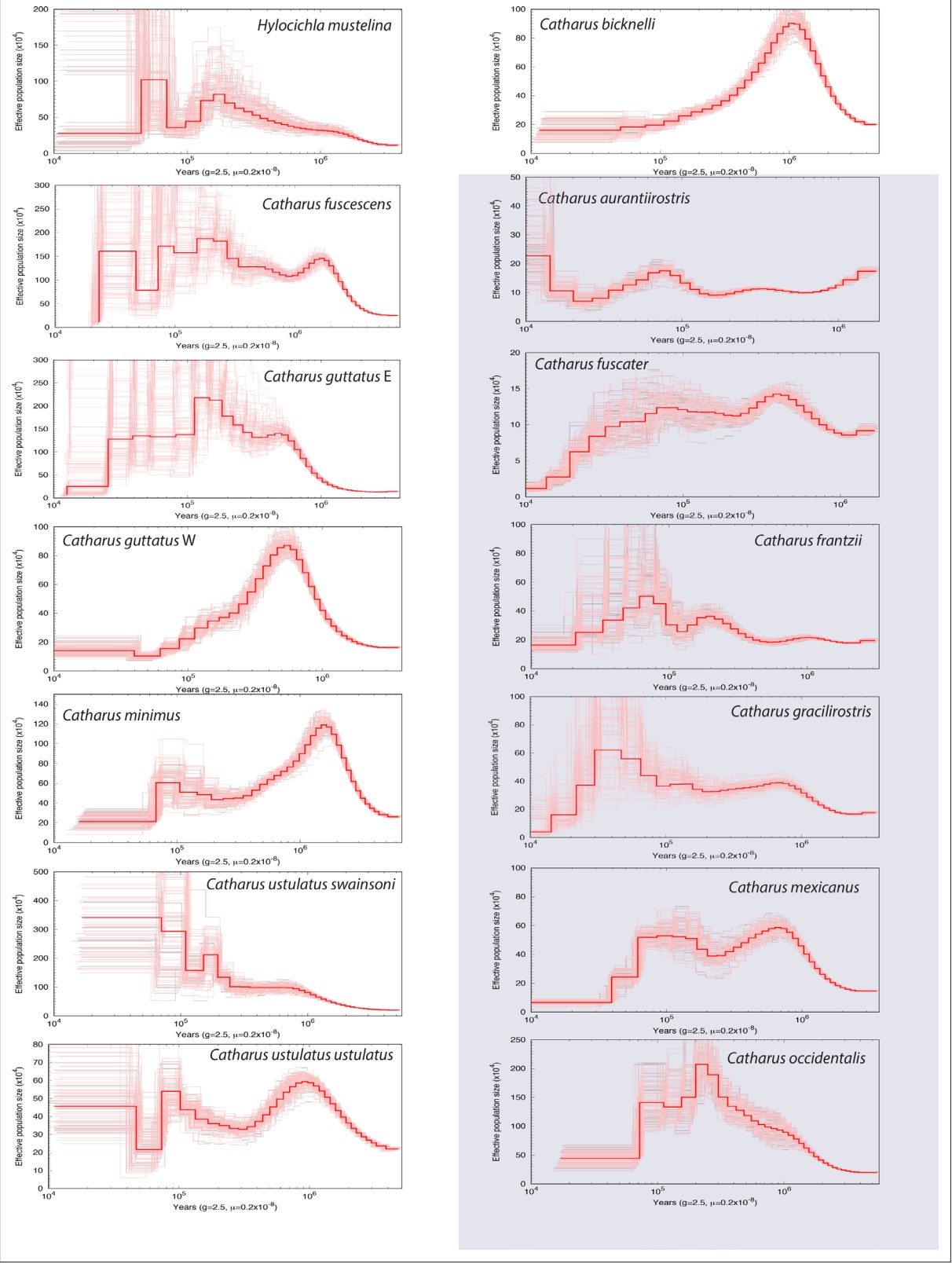

**Figure 5.** Montage of the historic effective population size curves of all lineages analyzed in this study, with each lineage in a separate panel (based on pairwise sequentially Markovian coalescent [PSMC] analyses). Sedentary lineages are highlighted in gray. Note that scales on both axes vary among panels, and that the *X* axis is on a log scale. Bold red lines are the main curves from the original data, and pink lines reflect 100 replicates from bootstrapped sequences. Bold red curves are all overlaid on common axes in the single panel of *Figure 4*.

usually show identical effective population size attributes at their origins, reflecting their shared histories (*Delmore et al., 2020*; *Li and Durbin, 2011*; *Nadachowska-Brzyska et al., 2016*). Inasmuch as many of our most interesting results are in deep time, where we expect current individuals of a lineage to have the most thorough mixing or amalgamation of lineage-specific metapopulation history, the effects of population structure are likely to become less important, and the results reflect (as we have necessarily interpreted) more of a lineage-wide phenomenon than a variably structured, demically biased one. However, it is clear that the latter bias becomes stronger as lineages progress toward the present, and for more recent times, population structure will be an important aspect of comparing effective population size estimates and their changes between migrant and resident lineages. In this respect, we were conservative in ignoring the most recent 50 Kyr.

It is also important to consider the time scales involved and the sampling regime. Glacial-interglacial cycles averaged ~100 Kyr back to 0.74 Mya and then averaged ~41 Kyr from then back to 2.47 Mya; about 50–60 of these cycles occurred (*Lisiecki and Raymo, 2005*: figure 4). This probably caused considerable levels of structuring and mixing in these lineages. In addition, in the PSMC output from one of our lineages, *C. ustulatus swainsonii*, we find that there are 54 time segments sampled for the Pleistocene, indicating the inadequacy of this method to reflect fine-scale changes and suggesting that each estimate is generally capturing a lot of both structuring and mixing.

At present, there is no way to fully disentangle the effects of population structure and geographic space on our results, but we make two observations leading us to infer that general differences in population structure are not the main factor driving our results. First, structure will have smaller effects on the PSMC signal in the deep past than in more recent history. Second, climatic changes are known to cause coordinated shifts among codistributed taxa, e.g., in promoting the partitioning of genetic variation among glacial refugia (*Hewitt, 2000*). A lack of coordinated shifts in our PSMC results suggests that climate-driven shifts in population structure are not a major feature (*Figure 5*).

## Geography and evolutionary ecology

Early population growth among migrant lineages is likely affected by some degree of ecological release as these birds engage with the novel environments their increased movements expose them to, and they occupy new niche space. However, in this system, we cannot readily decouple the expanded geographic and ecological opportunities that seasonal migration provides. The relationship between avian range size and migration is complex, but correlation with latitude is notably strong in North America (*Pegan and Winger, 2020*). Two aspects of our results suggest that geography operating alone is unlikely to explain the larger migrant population sizes observed (*Table 1*, *Figure 5*). The first is that early population growth is positive in all but one of the lineages we studied; despite a probable lack of marked geographic expansion among residents, they, too, experienced long-term growth. The second is that these initial growth periods are quite long, extending over much longer periods than we should expect for populations responding to opportunities for geographic expansion.

PSMC analyses excel at long time frames, but initial growth in residents and migrants suggests a strong role for ecological factors. Global climate reconstruction data do not suggest that there were similar long, steady trends during these time periods to match these thrush population size changes (*Lisiecki and Raymo, 2005*; *Figure 5*). Perhaps geographic space does make a difference, on average, between the two life-history strategies. That these growth periods are long also suggests the importance of ecological factors, though. For example, the boreal and subarctic woodland breeding habitats of most *C. ustulatus swainsonii* and *C. minimus* expanded dramatically across North America as glaciers receded since the last glacial maximum (LGM), a period of <20 Kyr (*Pielou, 1991*). A spatial response of such rapidity, less than 1% of the time of the initial historic growth phases of these two lineages (>3 Myr; *Table 1*), leads us to infer that breeding range size alone is not a satisfactory explanation for our results. This post-LGM expansion reflects census size, and not effective population size ($N_e$), but it implies that a migratory *Catharus* lineage's matching (or mapping) of $N_e$ to average long-term available geographic space is not likely to take millions of years. In other words, their comparatively recent rapid postglacial expansion into new space suggests a sort of preadaptation that is not apparent early in the lineages' histories. Together, these aspects of our results suggest that while geographic factors are involved, ecological factors are also important. Teasing these factors apart might also explain the finding of *Germain et al., 2023a*, that migrants in general have had larger populations than residents during the last 0.67 Myr.

Identifying key *ecological mechanisms* affecting population sizes among migrants is difficult, and they will likely vary among lineages. There are numerous factors potentially involved. Biotic interactions in general show greater effects on range limits at the warmer than the cooler edges (*Paquette and Hargreaves, 2021*). Long-distance migrants, especially in their extensive latitudinal movements, would receive a range expansion advantage in this respect regardless of continental landmass shape— and also through higher diel food harvest rates due to longer days and shorter nights. Many additional ecological factors affect the population sizes of migrants vs. residents, but the relationships between seasonal migration and predators, parasites, diseases, and competitors are extremely complex, and key aspects remain poorly understood (*Altizer et al., 2011*; *Newton, 2007*).

Seasonal migration can be integrally related to predator avoidance, but among birds, its effects are not understood in the full, circannual context relative to resident relatives, and they are not generalizable (*Brönmark et al., 2008*; *McKinnon et al., 2010*; *Newton, 2007*; *Rappole, 2013*). Exposure to parasites and pathogens varies both spatially and seasonally. Although breeding range diminishment of these exposures can be important in some migrants (probably not among all lineages), full annual exposure to both disease and parasite diversity is likely higher, resulting in at least some lineages showing larger immune defense organs (*Bennett and Fallis, 1960*; *Figuerola and Green, 2000*; *Jenkins et al., 2012*; *Møller and Erritzøe, 1998*; *Piersma, 1997*; *Ricklefs et al., 2017*). Exposure to higher circannual parasite and pathogen diversity might not be correlated with susceptibility, however; migration creates a geographic break that can lower transmission, causing migrants overall to be less vulnerable to population declines (*Hall et al., 2014*).

Presently, *C. minimus*, for example, shows an increased breeding and migratory prevalence and diversity of avian malaria, a blood parasite, but aspects of this relationship probably changed over the time scales involved (*Pulgarín-R et al., 2019*; *Ricklefs et al., 2014*). Avian migrants are more likely to be affected by blood parasites in ecological rather than evolutionary time, although the latter are also operating (*Hellgren et al., 2007*; *Ricklefs et al., 2014*). At the deeper times of avian families, seasonal migration is an important factor affecting blood parasite-host coevolution by breaking down the congruence of phylogenetic relationships between hosts and parasites, and there are also important differences among biting-fly vectors and their transmission of avian blood parasite hosts (*Jenkins et al., 2012*; *Ricklefs et al., 2014*). These issues suggest that reconstructing associations and their effects in deep time will be difficult.

Interspecific competition has long been studied between migrants and residents, but its potential role in affecting relative population sizes between the two groups is largely unknown (*Rappole, 2013*). As close relatives, these thrushes are likely to be each other's closest competitors, so it is conceivable that the increased relative abundance of migrants might depress realized population sizes among residents. In this group, geographic ranges overlap broadly during the nonbreeding season, but the tropical residents tend to occupy higher elevations than nonbreeding migrants (*Collar, 2005*). Both interspecific competition and inland, montane, and mature-forest retreats among older taxa are major factors in taxon cycles (*Ricklefs and Bermingham, 2002*; *Ricklefs and Cox, 1972*; *Wilson, 1961*) and might also be operating in this system.

## Conclusions

Paleodemographic analyses in a hypothesis-testing framework offer important new insights into the effects of the major life-history trait of seasonal migration on long-term effective population sizes. Our results indicate larger and more variable migrant populations relative to residents, and these populations grow in a way suggesting an important role not only for seasonal migration and geography, but also for evolutionary ecology. Although there are several plausible ecological mechanisms, their relative importance is unknown. The differences we found are between group averages, and variation among lineages includes residents that are migrant-like (*C. occidentalis*; *Table 1*, *Figures 1, 3, and 5*) and vice versa (e.g. *C. u. ustulatus*; *Table 1*, *Figure 5*). Understanding why residents like *C. occidentalis* show migrant-like historic population characteristics will help us understand the mechanisms affecting these lineages. Such reasons could be as simple as realized range expansion (i.e. geographic and ecological opportunity), and this hypothesis appears to fit the deeply split eastern-western sister lineages within species in which smaller western ranges have smaller effective population sizes (*C. guttatus* and *C. ustulatus*; *Figure 5*). Or the reasons might be as complex as a lineage dropping out of migration to become sedentary (*C. occidentalis* is sister to the migrant *C. guttatus*; *Figure 1*).

It is clear that seasonally migratory lineages in this system have, on average, fundamentally different effective population size attributes through evolutionary time than closely related resident lineages. Population structure is probably not a main driver of these results. What remains unclear are the relative roles of geographic and ecological expansions in causing these differences. Both are likely important. More studies in other systems will be needed to tease these roles apart and to determine which ecological mechanisms are involved.

What other patterns will emerge across geographic, temporal, and phylogenetic space?

Further sampling among other closely related groups containing mixed lineages of migrants and residents is needed, as are complementary approaches to reconstructing demographic history, using more individuals per lineage and other population genomic methods applicable to that type of sampling (e.g. *Beichman et al., 2017*; *Delmore et al., 2020*; *Marchi et al., 2021*). Such studies will also help determine the relative contributions of seasonal migration and its geographic and ecological expansion components in the context of what appears to be a normal tendency for early lineage growth (*Figures 1 and 5*, *Table 1*).

## Materials and methods

The thrush lineages in our study are from the genus *Catharus* and its sister, the single species in the genus *Hylocichla*. These comprise Nearctic-Neotropic seasonal migrants and Neotropical residents. Among the latter, some lineages (*C. aurantiirostris*, *occidentalis*, *frantzii*, and *mexicanus*) have some populations that exhibit some short-distance migration—likely partial migration or hard-weather movements, either in elevation or with extreme northern populations moving south (*Collar, 2005*). These lineages thus comprise two groups, which for simplicity we will refer to as 'seasonal migrants' and 'residents': seasonal migrants with extensive latitudinal movements, and residents that are either wholly sedentary or with some individuals moving rather limited distances within the Neotropics. The eight migrant lineages are: *H. mustelina*, *C. fuscescens*, *C. guttatus* E (eastern lineage), *C. guttatus* W (western lineage), *C. minimus*, *C. ustulatus swainsonii*, *C. u. ustulatus*, and *C. bicknelli*. (Note that four of these lineages represent two deeply divergent lineages, each within what are currently considered two biological species, *C. guttatus* and *C. ustulatus*; *Topp et al., 2013*.) The six resident lineages are: *C. aurantiirostris*, *C. fuscater*, *C. frantzii*, *C. gracilirostris*, *C. mexicanus*, and *C. occidentalis*. Over the full evolutionary history of this *Hylocichla-Catharus* clade, changes among lineages occurred in the trait of long-distance migration (*Outlaw et al., 2003*; *Voelker et al., 2013*; *Winker and Pruett, 2006*; *Figure 1*). Our study does not assume that this trait was constant within each lineage after the speciation events when these switches occurred. Instead, we consider that for a major trait like long-distance migration, losses and gains at the full lineage level are likely to be infrequent, and that current evidence of trait distribution on a phylogeny provides a powerful comparative framework in which to elucidate the effects of that trait on other lineage attributes.

DNA was extracted from high-quality voucher specimens or blood samples for 14 thrush lineages (*Appendix 1—table 1*; data are available on NCBI-SRA, see *Delmore and Winker, 2024*). Whole-genome sequencing libraries were constructed using Nextera (Illumina) DNA Flex Library Prep kits, and libraries were sequenced on a NovaSeq S4 (Illumina, San Diego, CA, USA) using paired-end 150 bp reads. Reads were aligned to the *C. ustulatus* genome (inland subspecies, Reference bCatUst1, NCBI PRJNA613294) using bwa (mem algorithm with default settings, *Li and Durbin, 2009*). We did not use a maskfile. Identical treatment of all lineages in these respects should provide a strong foundation for a comparative study like this among close relatives. Resulting .sam files were converted to .bam format with samtools (*Li et al., 2009*); picardtools was used to clean, sort, and add read groups to .bam files (https://broadinstitute.github.io/picard/; RRID:SCR_006525). Between 97% and 98% of reads mapped to the reference for an average read depth of 23× (range 15.9–29.8×).

We analyzed whole-genomic data from the .bam files using the PSMC approach in the software package psmc (*Li and Durbin, 2011*; our code is available at doi 10.6084/m9.figshare.28828478). This analysis uses pairwise (allelic) estimates of coalescent times for each locus across the genome in a nonoverlapping manner, developing a distribution of times to most recent common ancestor, and the frequency of these coalescent events is inversely proportional to effective population size (*Li and Durbin, 2011*; *Mather et al., 2020*).

We used a generation time of 2.5 year, taken as an average among many of these species (*Dellinger et al., 2020*; *Evans et al., 2020*; *Mack and Yong, 2020*; *Saether et al., 2005*; *Townsend et al.,*

*2020*; *Whitaker et al., 2020*), and a mutation rate of $2.3\times10^{-9}$ mutations per site per year (*Smeds et al., 2016*). Single individuals represent lineage characteristics using this approach (*Li and Durbin, 2011*; *Mazet et al., 2015*; *Nadachowska-Brzyska et al., 2015*). Importantly for our focal questions, variations in these parameters do not affect the shape of the curve of population sizes ($N_e$) through time, but rather the values of the time and $N_e$ estimates (*Nadachowska-Brzyska et al., 2015*). Being closely related and of similar size, we expect these taxa to have very similar generation times and mutation rates (e.g. *Bird et al., 2020*; *Healy et al., 2014*). However, seasonal migration can have an effect on survival and thus generation time, and *Bird et al., 2020*, produced modeled estimates for this group, providing an alternate approach. The basis for these modeled values is tiny, and the accuracy of the modeled results is uncertain, so we use these estimates with caution. But as a check on our results, we ran all of our analyses again with the variable generation times given by *Bird et al., 2020*. As expected, most of the differences are time-related, and, importantly, the overall results are not different (*Appendix 1—table 2*).

*Hilgers et al., 2024*, recently showed that artifactual peaks can occur due to recent time parameter settings in the PSMC program (*Li and Durbin, 2011*) that are often used as a default setting. They recommended that such recent peaks can be eliminated by breaking the first atomic time interval (typically 4) into either two (2+2) or four (1+1+1+1) smaller windows. We had four lineages that appeared to have such peaks: *H. mustelina*, *C. fuscescens*, *C. guttatus* E, and *C. gracilirostris*. For the latter three, breaking the recent period parameter into four windows (1+1+1+1) eliminated these peaks. For *H. mustelina*, smaller windows (2+2 and 1+1+1+1) made the recent peak even higher, so we have used the default setting (the most conservative result), as we did with the remaining taxa that did not show such peaks.

The PSMC method is not foolproof; e.g., aspects of selection, inbreeding, and isolation can affect population size estimates (*Mather et al., 2020*). No method of genomic population demographic reconstruction is known to be consistently and reliably accurate (*Marchi et al., 2021*), but uniform application of a powerful method like PSMC to genome-scale data in a closely related group of lineages can provide new insights into the evolutionary effects of major life-history strategies.

For all variables, PSMC output for times more recent than 50 Kya was ignored, because recent estimates of effective population size using this method are less reliable (*Li and Durbin, 2011*;

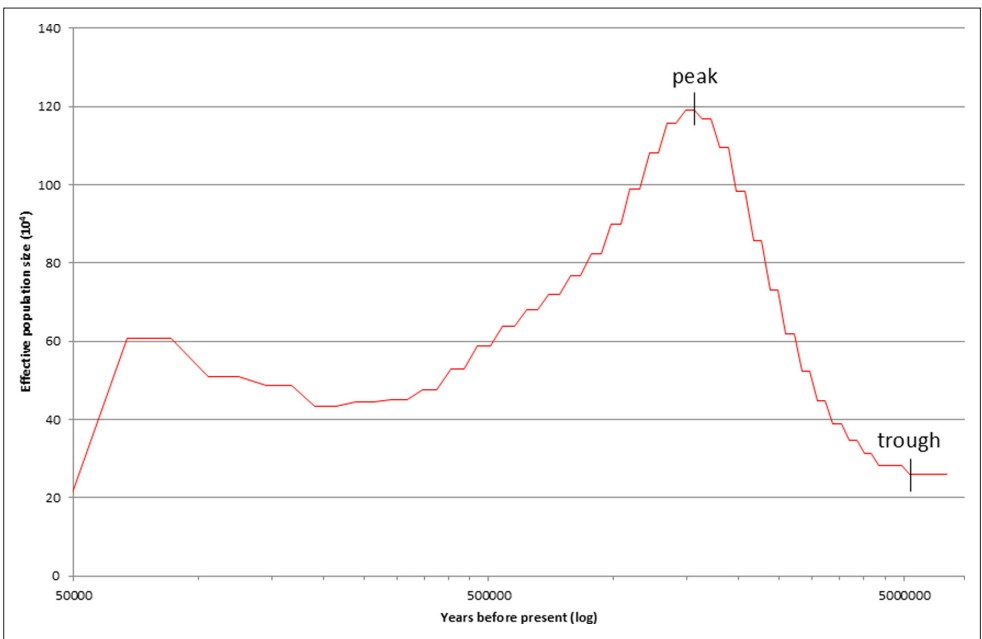

**Figure 6.** Graphic presentation of the *C. minimus* pairwise sequentially Markovian coalescent (PSMC) dataset, showing effective population size ($N_e\times10^4$) from 50 Kyr back in time to the lineages' origins as estimated from genomic data. The variables in our analyses are the mean and SD of the effective population size ($N_e$) values, *deltaT* of initial growth ($\text{time}_{\text{trough}} - \text{time}_{\text{peak}}$), the degree of that growth ($1 - [N_{\text{trough}}/N_{\text{peak}}]$), and the rate of that growth (degree/*deltaT*).

*Nadachowska-Brzyska et al., 2016*). We consider this to be a conservative cutoff date. Our two focal variables, mean effective population size ($N_e$) and the coefficient of variation (SD/mean), were obtained from standard PSMC output. Our examination of early population growth involved deriving two additional variables from the PSMC output: degree of initial growth ($1 - (N_{trough}/N_{peak})$), and rate of growth (degree/time period over which initial growth occurred; the latter we term *deltaT*; *Figure 6*).

We note that estimates of the dates of divergence events in *Catharus* evolution vary by study, genetic marker, methodology, and parameter estimates such as generation time and mutation or substitution rates (*Outlaw et al., 2003*; *Topp et al., 2013*; *Voelker et al., 2013*; *Winker and Pruett, 2006*). Date estimates in our study that differ from previous work reflect these methodological issues, and here it is the relative values among lineages within our study that are most important.

We tested the assumption of phylogenetic independence of continuous characters obtained from the PSMC analyses using a phylogeny reconstructed using maximum likelihood from 1.5 Mb of autosomal sequence from chromosome 22 in MEGA X (*Kumar et al., 2018*; *Figure 1*), which produced the expected topology given prior work (e.g. *Everson et al., 2019*), and, for phylogenetic independence, the Abouheif-Moran approach (*Abouheif, 1999*; *Münkemüller et al., 2012*) as implemented in the R package adephylo (*Jombart et al., 2010*). For those variables not showing phylogenetic signal, we tested our hypotheses using a one-tailed Mann-Whitney *U*-test. This nonparametric test has the advantage of not requiring that particular assumptions are met (e.g. homogeneity of variance) nor precise measurements (thus accommodating uncertainty in our $N_e$ estimates; *Sokal and Rohlf, 1995*; *Whitlock and Schluter, 2015*).

## Acknowledgements

We thank the Field Museum of Natural History, the Museum of Southwestern Biology, the Louisiana State University Museum of Natural Science, and the University of Alaska Museum for tissue loans. Fern Spaulding, Symcha Gillette, and Andrew Hillhouse kindly assisted with lab work. John Rappole provided helpful correspondence, and comments from three anonymous reviewers helped us improve the manuscript. We also thank Mark Balman and BirdLife International for the data used in creating the distribution maps in *Figure 2*. The Kessel Fund (University of Alaska Fairbanks), Texas A&M University, and the U.S. National Science Foundation (IOS-2143004) provided support for this work.

## Additional information

### Funding

| Funder | Grant reference number | Author |
| --- | --- | --- |
| Kessel Fund | | Kevin Winker |
| Texas A&M University | | Kira Delmore |
| National Science Foundation | IOS-2143004 | Kira Delmore |

The funders had no role in study design, data collection and interpretation, or the decision to submit the work for publication.

### Author contributions
Kevin Winker, Kira Delmore, Conceptualization, Formal analysis, Funding acquisition, Validation, Investigation, Visualization, Methodology, Writing – original draft, Writing – review and editing

### Author ORCIDs
Kevin Winker ⬤ https://orcid.org/0000-0002-8985-8104
Kira Delmore ⬤ https://orcid.org/0000-0003-4108-9729

Joint Public Review: https://doi.org/10.7554/eLife.90848.3.sa1
Author response https://doi.org/10.7554/eLife.90848.3.sa2

## Additional files

### Supplementary files
MDAR checklist

Supplementary file 1. Specimen data, NCBI-SRA numbers, and alternative generation time PSMC results.

### Data availability

Data are available on NCBI-SRA at accession numbers PRJNA1112856 and PRJNA979932.

The following datasets were generated:

| Author(s) | Year | Dataset title | Dataset URL | Database and Identifier |
|-----------|------|---------------|-------------|-------------------------|
| Delmore K, Winker K | 2024 | Data accompanying phylogenomic analysis of the Catharus genus | https://www.ncbi.nlm.nih.gov/bioproject/?term=PRJNA1112856 | NCBI BioProject, PRJNA1112856 |
| Delmore K, Winker K | 2023 | Mapping seasonal migration in a songbird hybrid zone | https://www.ncbi.nlm.nih.gov/bioproject/?term=PRJNA979932 | NCBI BioProject, PRJNA979932 |

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

# Appendix 1

**Appendix 1—table 1.** Specimen data and NCBI-SRA numbers (PRJNA1112856).
Vouchered specimens are housed in the following institutions: UAM (University of Alaska Museum), MSB (Museum of Southwestern Biology, University of New Mexico), LSUMNS (Louisiana State University Museum of Natural Science), and FMNH (Field Museum of Natural History).

| Institution | Catalog # | Species | Year | Field no. | Locality | NCBI-SRA |
|---|---|---|---|---|---|---|
| UAM | 28620 | *H. mustelina* | 2010 | KSW5403 | Belize: Toledo District; Big Falls | SRR29089747 |
| UAM | 27774 | *C. fuscescens* | 2007 | KSW5151 | Belize: Toledo District; Big Falls | SRR29089742 |
| UAM | 15202 | *C. guttatus E* | 1992 | KSW4013 | USA: Vermont; Brandon | SRR29089748 |
| UAM | 26337 | *C. guttatus W* | 2008 | UAMX5095 | USA: Alaska; Kodiak | SRR29089739 |
| UAM | 22642 | *C. minimus* | 2003 | KSW5000 | USA: Alaska; Fairbanks | SRR29089741 |
| n.a. (blood) | KF15K01 | *C. ustulatus swainsonii* | 2011 | KF15K01 | Canada: British Columbia, Kamloops | Pending |
| n.a. (blood) | KF01K01 | *C. u. ustulatus* | 2011 | KF01K01 | Canada: British Columbia, Kamloops | SRS18060177 |
| UAM | 19996 | *C. bicknelli* | 2000 | KSW3633 | USA: Vermont; Mt Mansfield | SRR29089740 |
| UAM | 25341 | *C. aurantiirostris* | 2004 | MJM1154 | Panama: Chiriqui; El Salto | SRR29089749 |
| MSB | 31939 | *C. fuscater* | 2008 | MSB31939 | Peru: Amazonas; 4.5 km N Tullanya | SRR29089746 |
| UAM | 25098 | *C. frantzii* | 2004 | KSW4485 | Panama: Chiriqui; Volcan Baru | SRR29089744 |
| LSUMNS | 138784 | *C. gracilirostris* | 1990 | JMB1065; B-16270 | Costa Rica: San Jose; Cerro de la Muerte | SRR29089750 |
| UAM | 10352 | *C. mexicanus* | 1994 | PEP2489 | Mexico: Veracruz; Volcan San Martin | SRR29089743 |
| FMNH | 343305 | *C. occidentalis* | 1989 | MEX408 | Mexico: Oaxaca; Totontepec | SRR29089745 |

**Appendix 1—table 2.** Data from pairwise sequentially Markovian coalescent (PSMC) analyses reflecting effective population sizes ($N_e$) ($\times 10^4$) through history at depths >50 Kyr (using variable generation times) and the five variables derived and analyzed from that output.
Taxa shaded in gray are Neotropical residents.

| Taxon | Mean $N_e$ ($\pm$SD) | SD/mean | Degree of early growth 1 - ($N_{trough}/N_{peak}$) | Rate of early growth degree/deltaT | deltaT |
|---|---|---|---|---|---|
| *H. mustelina* | 35.63 ($\pm$20.11) | 0.56 | 0.86 | 3.03E-07 | 2,834,197 |
| *C. fuscescens* | 95.77 ($\pm$56.07) | 0.58 | 0.83 | 1.79E-07 | 4,627,125 |
| *C. guttatus E* | 78.69 ($\pm$76.09) | 0.97 | 0.90 | 4.89E-07 | 1,850,522 |
| *C. guttatus W* | 40.15 ($\pm$24.11) | 0.60 | 0.81 | 3.59E-07 | 2,266,059 |
| *C. minimus* | 63.71 ($\pm$28.83) | 0.45 | 0.78 | 2.35E-07 | 3,323,013 |
| *C. ustulatus swainsonii* | 76.61 ($\pm$66.61) | 0.87 | 0.79 | 2.12E-07 | 3,740,764 |
| *C. ustulatus ustulatus* | 39.56 ($\pm$12.15) | 0.31 | 0.63 | 2.16E-07 | 2,909,628 |
| *C. bicknelli* | 46.90 ($\pm$24.12) | 0.51 | 0.78 | 3.01E-07 | 2,580,641 |
| *C. aurantiirostris* | 12.39 ($\pm$2.78) | 0.22 | –0.75 | –7.93E-07 | 942,318 |
| *C. fuscater* | 11.33 ($\pm$1.66) | 0.15 | 0.39 | 5.57E-07 | 713,817 |
| *C. frantzii* | 24.97 ($\pm$8.61) | 0.34 | 0.17 | 1.93E-07 | 886,294 |

*Appendix 1—table 2 Continued on next page*

Appendix 1—table 2 Continued

| Taxon | Mean $N_e$ (±SD) | SD/mean | Degree of early growth 1 - ($N_{trough}/N_{peak}$) | Rate of early growth degree/deltaT | deltaT |
|---|---|---|---|---|---|
| C. gracilirostris | 30.80 (±11.08) | 0.36 | 0.58 | 3.25E-07 | 1,775,757 |
| C. mexicanus | 37.10 (±16.18) | 0.44 | 0.75 | 2.92E-07 | 2,575,082 |
| C. occidentalis | 76.39 (±53.14) | 0.70 | 0.91 | 1.97E-07 | 4,595,855 |
| Means (±SD) | | | | | |
| Migrants | 59.63 (±38.48)† | 0.61 (±0.21)* | 0.798 (±0.08)* | 2.87E-7 (±1.01E-7) | 3,016,494 (±875,860)* |
| Residents | 32.16 (±15.58) | 0.36 (±0.19) | 0.342 (±0.59) | 1.284E-7 (±4.71E-7) | 1,914,854 (+1,489,246) |

**p<0.05.

†p<0.01.

