## [Editor Report · eLife Assessment]

This is a **valuable** study of the role that life history differences might play in determining population size and demography. While concerns about generation times and population structure leave the evidence for the claims in parts **incomplete**, the work is of considerable interest to anyone who tries to understand evolutionary consequences of life history changes.

---

## [Referee Report · Joint Public Review]

Summary:

This interesting study applies the PSMC model to a set of new genome sequences for migratory and nonmigratory thrushes and seeks to describe differences in the population size history among these groups. The authors create a set of summary statistics describing the PSMC traces - mean and standard deviation of Ne, plus a set of metrics describing the shape of the oldest Ne peak - and use these to compare across migratory and resident species (taking single samples sequenced here as representative of the species). The analyses are framed as supporting or refuting aspects of a biogeographic model describing colonization dynamics from tropical to temperate North and South America.

Strengths:

* This is a creative use of PSMC to test explicit a priori hypotheses about season migration and Ne. The PSMC analyses seem well done and the authors acknowledge much of the complexity of interpretation in the discussion.

* We appreciate the test-of-hypothesis design of the study and the explicit formulation of three main expectations to test. The data analysis has been done with appropriate available tools.

Key weaknesses from the original round of review:

* Short of developing some novel theory deep in the PSMC model, I think readers would need to see simulations showing that the analyses employed in this paper are capable of supporting or refuting their biogeographic hypothesis before viewing them as strongly supporting a specific biogeographic model. Tools like msprime and stdpopsim can be used to simulate genome-scale data with fairly complex biogeographic models. Running simulations of a thrush-like population under different biogeographic scenarios and then using PSMC to differentiate those patterns would be a more convincing argument for the biogeographic aspects of this paper. The other benefit of this approach would be to nail down a specific quantitative version of the taxon cycles model referenced in the abstract, and it would allow the authors to better study and explain the motivation behind the specific summary statistics they develop for PSMC posthoc analysis.

* The authors hypothesized that the wider realized breeding and ecological range characterising migrants versus resident lineages could be a major drive for increased effective population size and population expansion in migrants versus residents. I understand that this pattern (wider range in migrants) is a common characteristic across bird lineages and that it is viewed as a result of adapting to migration. A problem that I see in their dataset is that the breeding grounds range of the two groups are located in very different geographic areas (mainly South versus North America). The authors could have expanded their dataset to include species whose breeding grounds are from the two areas, regardless of their migratory behaviour, as a comparison to disentangle whether ecological differences of these two areas can affect the population sizes or growth rates.

* As I understand from previous literature, the time-scale to population growth and estimates of effective population sizes considered in the present paper for the resident versus migratory clades seem to widely predate the times to speciation for the same lineages, which were reported in previous work of the same authors (Everson et al 2019) and others (Termignoni-Garcia et al 2022). This piece of information makes the calculation of species-specific population size changes difficult to interpret in the light of lineages' comparison. It is unclear what the authors consider to be lineage-specific in these estimates, as the clades were likely undergoing substantial admixture during the time predating full isolation.

* Regarding the methodological difficulties in interpreting the impact of population structure on the estimates of effective population sizes with the PSMC approach, I would think that performing simulations to compare different scenarios of different degrees of structured populations would have helped substantially understand some of the outcomes.

* The authors use an average generation time for all taxa, but the citations imply generation time is known for at least some of them. Are there differences in generation time associated with migration? I am not a bird biologist, but quick googling suggests maybe this is the case? (https://doi.org/10.1111/1365-2656.13983). I think it important the authors address this, as differences in generation time I believe should affect estimates of Ne and growth.

[Editors' note: the original reviews in full are here: https://elifesciences.org/reviewed-preprints/90848/reviews. The reviewers were not available to comment on the latest version of the submission.]

---

## [Author Response]

The following is the authors’ response to the original reviews.

**eLife Assessment**
This study presents valuable finding regarding the role of life history differences in determining population size and demography. The evidence for the claims is still partially incomplete, with concerns about generation times and population structure. Nonetheless, the work will be of considerable interest to biologists thinking about the evolutionary consequences of life history changes.

Thank you. We have addressed the generation time and population structure issues in detail in our revision and hope that you, like us, find them to be of sufficiently low concern (i.e., they are not driving the results) that they do not overshadow the main findings and conclusions.

The opportunity to make in-depth revisions also helped the manuscript in two ways unanticipated by both us and the reviewers. First, KW made a mistake in the original analysis of phylogenetic signal, and catching that error simplifies that aspect of the study (there is none in our measured variables). Second, in June 2024 Hilgers et al. (2024; https://doi.org/10.1101/2024.06.17.599025) posted an important manuscript to *bioRxiv* noting the possibility of false population size peaks in PSMC analyses using the standard default settings. Our results had three of those, which we have eliminated. *Ne*ither of these issues affect the overall conclusions, but their resolution improves the work.

**Public Reviews:**

**Reviewer #1 (Public Review):**
Summary:This interesting study applies the PSMC model to a set of new genome sequences for migratory and nonmigratory thrushes and seeks to describe differences in the population size history among these groups. The authors create a set of summary statistics describing the PSMC traces - mean and standard deviation of *Ne*, plus a set of metrics describing the shape of the oldest *Ne* peak - and use these to compare across migratory and resident species (taking single samples sequenced here as representative of the species). The analyses are framed as supporting or refuting aspects of a biogeographic model describing colonization dynamics from tropical to temperate North and South America.Strengths:At a technical level, the sequencing and analysis up through PSMC looks good and the paper is engaging and interesting to read as an introduction to some verbal biogeographic models of avian evolution in the Pleistocene.

The core findings - higher and more variable *Ne* in migratory species - seem robust, and the biogeographic explanation is plausible.

Thanks. We thought so as well. Our analyses go beyond being simply descriptive and test some simple hypotheses, including a biogeographic+ecological expansion opportunity gained in some lineages through the adoption of a seasonal migration life-history strategy.

Weaknesses:I did not find the analyses particularly persuasive in linking specific aspects of clade-level PSMC patterns causally to evolutionary driving forces. To their credit, the authors have anticipated my main criticism in the discussion. This is that variation in population size inferred by methods like PSMC is in "effective" terms, and the link between effective and census population size is a morass of bias introduced by population structure and selection so robustly connecting specific aspects of PSMC traces to causal evolutionary forces is somewhere between extremely difficult and impossible.

As R1 notes, we do not attempt to link effective population sizes and census sizes (though we do discuss this), and we are also careful to discuss correlated rather than causative factors when going beyond the overarching hypotheses regarding life-history strategy.

Population structure is the most obvious force that can generate large *Ne* changes mimicking the census-sizefocused patterns the authors discuss. The authors argue in the discussion that since they focus on relatively deep time (>50kya at least, with most analyses focusing on the 5mya - 500kya range) population structure is "likely to become less important", and the resident species are usually more structured today (true) which might bias the findings against the observed higher *Ne* in migrants.

To clarify, the patterns we discuss are entirely related to effective population size, not census size. But, yes, this is why we’ve given population structure its own section in the Discussion.

But is structure really unimportant in driving PSMC results at these specific timescales? There is no numerical analysis presented to support the claim in this paper. The biogeographic model of increased temperate-latitude land area supporting higher populations could yield high *Ne* via high census size, but shifts in population structure (for example, from one large panmictic population to a series of isolated refugial populations as a result of glaciation-linked climate changes) could plausibly create elevated and more variable *Ne*. Is it more land area and ecological release leading to a bigger and faster initial *Ne* bump, or is it changes in population connectivity over time at expanding range edges, or is the whole single-bump PSMC trace an artifact of the dataset size, or what? The authors have convinced me that the *Ne* history of migratory thrushes is on average very different from nonmigrant thrushes, but beyond that it's unclear what exactly we've learned here about the underlying process.

We do not argue that population structure is unimportant, only that it is less important as one goes into deeper time. Further, we agree with the reviewer’s observation above that structure is more likely to bias nonmigrant estimates of *Ne*. In other words, following Li & Durbin’s (2011) simulations, we interpret that an inflated *Ne* due to structure should occur more often among residents. We have clarified this in the revision. We also agree that what we’ve learned about the underlying process is not entirely clear, but as we stated, population structure does not seem to be the main driver, and there is evidence that both biogeographic and ecological factors are involved. With this being the first time that these questions have been asked, we think we’ve made an important advance and that we’ve opened a number of avenues for future study.

It also important to consider the time scales involved and the sampling regime. Glacial-interglacial cycles averaged ~100 Kyr back to 0.74 Mya and then averaged ~41 Kyr from then back to 2.47 Mya; about 50-60 of these cycles occurred (Lisiecki & Raymo 2005: fig. 4). This probably caused a lot of population structuring and mixing in these lineages. In addition, in the PSMC output from one of our lineages, *C. ustulatus swainsonii*, we find that there are 54 time segments sampled for the Pleistocene, indicating the inadequacy of this method to reflect fine-scale changes and suggesting that each estimate is capturing a lot of both phenomena, structuring and mixing. We have added this to the revision.

I generally agree with the authors that "at present there is no way to fully disentangle the effects of population structure and geographic space on our results". But given that, I think there are two options - either we can fully acknowledge that oversimplified demographic models like PSMC cannot be interpreted as supporting evidence of any particular mechanistic or biogeographic hypothesis and stop trying to use them to do that, or we have to do our best to understand specifically which models can be distinguished by the analyses we're employing.Short of developing some novel theory deep in the PSMC model, I think readers would need to see simulations showing that the analyses employed in this paper are capable of supporting or refuting their biogeographic hypothesis before viewing them as strongly supporting a specific biogeographic model. Tools like msprime and stdpopsim can be used to simulate genome-scale data with fairly complex biogeographic models. Running simulations of a thrush-like population under different biogeographic scenarios and then using PSMC to differentiate those patterns would be a more convincing argument for the biogeographic aspects of this paper. The other benefit of this approach would be to nail down a specific quantitative version of the taxon cycles model referenced in the abstract, and it would allow the authors to better study and explain the motivation behind the specific summary statistics they develop for PSMC posthoc analysis.

These could very well be fruitful pursuits for future work, but they are beyond the scope of this paper. The impossibility of reconstructing ranges through deep time makes anything other than the very general biogeographic hypothesis we’ve posed an uncertain pursuit. Also, a purely biogeographic approach neglects the likelihood of ecological expansion also being involved. We get at the importance of the latter in the “Geography and evolutionary ecology” section of the Discussion. Below, the editor states that discussions among reviewers indicate that simulations are not warranted at this time. We agree that the complexities involved are substantial, to the point of making direct relevance to this empirical study uncertain (especially in such an among-lineage context). Regarding taxon cycles, we merely point out that that conceptual framework seems relevant given our findings. This was not even remotely anticipated at the outset of the study, so we are reluctant to do anything more than point out its possible relevance in several aspects of the results. Finally, the motivation for the study’s summary statistics were entirely driven by the hypotheses, as given in Methods, and due to an earlier error (noted above), there are no post-hoc analyses in the revision. Sorry for the needless confusion.

**Reviewer #2 (Public Review):**
Summary:Winker and Delmore present a study on the demographic consequences of migratory versus resident behavior by contrasting the evolutionary history of lineages within the same songbird group (thrushes of the genus Catharus).Strengths:I appreciate the test-of-hypothesis design of the study and the explicit formulation of three main expectations to test. The data analysis has been done with appropriate available tools.Weaknesses:The current version of the paper, with the case study chosen, the results, and the relative discussion, is not satisfying enough to support or reject the hypotheses here considered.

Given the stated strengths, the weaknesses noted seem a little incongruous, but we understand from the comments below that the reviewer would like to see the study redesigned and expanded.

The authors hypothesized that the wider realized breeding and ecological range characterising migrants versus resident lineages could be a major drive for increased effective population size and population expansion in migrants versus residents. I understand that this pattern (wider range in migrants) is a common characteristic across bird lineages and that it is viewed as a result of adapting to migration. A problem that I see in their dataset is that the breeding grounds range of the two groups are located in very different geographic areas (mainly South versus North America). The authors could have expanded their dataset to include species whose breeding grounds are from the two areas, regardless of their migratory behaviour, as a comparison to disentangle whether ecological differences of these two areas can affect the population sizes or growth rates.

Because the questions are about the migratory life history strategy and the best way to get at this is in a phylogenetic framework, we’re not sure how we could effectively add species “regardless of their migratory behavior.” Further, we know that migration causes lineages to experience variable ecological conditions that include breeding, migration, and wintering conditions. Obligate migrants are going to have different breeding ranges from their close relatives, and the more distantly related species are, the less likely it is that they respond to particular ecological conditions the same way. So we do not think that an approach that included miscellaneous species from northern and southern regions would strengthen this study. Here, the comparative framework of closely related lineages that possess or lack the trait of interest is a study design strength. We do agree, however, that future work is needed that does encompass more lineages (we would argue in a phylogenetic context), and that disentangling the effects of geography and ecology will also be an important future endeavor.

As I understand from previous literature, the time-scale to population growth and estimates of effective population sizes considered in the present paper for the resident versus migratory clades seem to widely predate the times to speciation for the same lineages, which were reported in previous work of the same authors (Everson et al 2019) and others (Termignoni-Garcia et al 2022). This piece of information makes the calculation of species-specific population size changes difficult to interpret in the light of lineages' comparison. It is unclear what the authors consider to be lineage-specific in these estimates, as the clades were likely undergoing substantial admixture during the time predating full isolation.

We do recognize that timing estimates vary among studies. Differences among studies in important variables like markers, methods, generation time, and mutation or substitution rates create much of this uncertainty. Also, we are not confident in prior dating efforts in this group, largely because of gene flow and its effects on bringing estimates closer to the present. As we point out (line 485), differences among studies on these issues do not detract from the strengths here for within-study, among-lineage contrasts. In short, the timing could be off in an among-study context (and likely is with prior work, given gene flow), but relative performance of among-lineage *Ne* differences is less susceptible to these factors. This was shown fairly well in Li & Durbin’s initial use of the method among human populations. Regarding substantial admixture, PSMC curves often unite at their origins with sister lineages (when they were the same lineage). A good example is with the two *C. guttatus* E & W curves in Fig. S3, which still have substantial gene flow today (they are subspecies and in contact), yet they show remarkably different *Ne* curves through their history. It is not possible to mark a cutoff point for each lineage that represents the cessation of admixture with another lineage (e.g., Everson et al. 2019 showed substantial admixture between three full species in this group); that period can be very long (Price et al. 2008), varies among lineages, and will not be available for deeper lineage divergences in the phylogeny. We therefore chose to use all of the time intervals retrievable from the genomic data in each lineage, considering that this uniform treatment is the best approach for our among-lineage comparison. And note that we were careful to label these as “the lineages’ PSMC inception” (line 190).

Regarding the methodological difficulties in interpreting the impact of population structure on the estimates of effective population sizes with the PSMC approach, I would think that performing simulations to compare different scenarios of different degrees of structured populations would have helped substantially understand some of the outcomes.

The complexities of such modeling in a system like this are daunting. The different degrees of structuring among all of these lineages across just a single glacial-interglacial cycle would necessitate a lot of guesswork; projecting that back across 50-60 such cycles just in the Pleistocene would probably end up being fiction. Disentangling the effects of structure versus changes in *Ne* in a system like this would probably not be possible with that approach and these data. As noted above and below, there was agreement among reviewers and the editor that simulations in this case are not warranted for revision. We have added the nature of the glacialinterglacial cycles and the PSMC sampling time segments to help readers understand this better (see above in response to R1, and lines 272-278).

Additionally, I have struggled to understand if migratory behaviour in birds is considered to be acquired to relieve species competition, or as a consequence of expanded range (i.e., birds expand their range but their feeding ground is kept where speciation occurred as to exploit a ground with higher quality and abundance of seasonal local resources).

The origins of migration have been a struggle for researchers since the subject was taken up. But how the trait was acquired among these species does not really matter for our study. Here, migratory lineages possess different biogeographic+ecological attributes than their close relatives that are sedentary. Our focus is on the presence and absence of this life-history trait.

The points raised above could be considered to improve the current version of the paper.

Thank you. We appreciate the opportunity to guide our revision using your comments.

**Reviewer #3 (Public Review):**
Summary:This paper applies PSMC and genomic data to test interesting questions about how life history changes impact long-term population sizes.Strengths:This is a creative use of PSMC to test explicit a priori hypotheses about season migration and *Ne*. The PSMC analyses seem well done and the authors acknowledge much of the complexity of interpretation in the discussion.Weaknesses:The authors use an average generation time for all taxa, but the citations imply generation time is known for at least some of them. Are there differences in generation time associated with migration? I am not a bird biologist, but quick googling suggests maybe this is the case (https://doi.org/10.1111/1365-2656.13983). I think it important the authors address this, as differences in generation time I believe should affect estimates of *Ne* and growth.

Good point. The study cited by the reviewer encompasses a much higher degree of variation in body size and thus generation time. Differences in generation time in similarly sized close relatives, as in our study, should be small, and our approach has been to average those that are known. Unfortunately, generation times are not known for all of these species, but given their similarity in size we can have reasonable confidence in their being similar. We used data from the life-history research available (as cited) to obtain our average; there are not appropriate data for the residents, though. However, there is thought to be a generation time cost to seasonal migration in birds, and Bird et al. (2020) included this in their estimates to provide modeled values for all of the lineages we studied. We’re leery of using modeled values where good data for the nonmigrants in this group don’t exist (and the basis for quantifying this cost is tiny), but we recognize that this second approach is available and could leave some doubt in our results if not pursued. So we re-did everything with the modeled generation times of Bird et al. (2020). As expected, most of the differences are time-related. Importantly, our overall results are not different. We present them as Table S2 and have added the details on this to the Methods.

The writing could be improved, both in the introduction for readers not familiar with the system and in the clarity and focus of the discussion.

We have added a phylogeny (new Fig. 1) to help readers better understand the system, and we’ve re-worked the Discussion to make it clearer what is clarified by our results and what remains unclear.

**Recommendations for the authors:**

**Reviewing Editor comment:**
I note that discussion among the reviewers made clear that simulations are probably not the right answer given the complexity of the modeling required.

We appreciate this conclusion, with which we agree.

**Reviewer #2 (Recommendations For The Authors):**
Apologies for the delay with the review, which came at a very busy time. I hope you will find my comments helpful.

Thanks. Your comments are helpful, and we fully understand how reviews (and our revisions!) have to wait until more pressing needs are addressed.

I enjoyed reading the manuscript but I believe that the discussion sections could be heavily rewritten for better clarity. The discussion is sometimes redundant and lacks some flow/clarity. In a nutshell, I had the feeling that a bit of everything is thrown in the discussion but clear conclusions are not made.

Yes, the Discussion has been difficult to write, because more issues arose in the Results than we anticipated at the outset. We feel that discussing them is relevant, but we agree that much remains unclear. This coupling of paleodemographics with geography and ecology is a new area, which opens some important new (and relevant) areas to consider. So clarity is not possible in some areas. We’ve revised to point out where we do have clarity (e.g., in migrant lineages having different paleodemographic attributes than nonmigrants) and where only further study can provide clarity (e.g., in the roles of geography versus ecology). The journal format does not seem to have secondary subheaders, but we’ve used bold in one place to highlight ‘ecological mechanisms’ to offset that section, one of the more complex. We’ve also added a paragraph in the conclusions to clarify where we have clear takeaways and where uncertainties remain.

**Reviewer #3 (Recommendations For The Authors):**
The introduction should engage the reader with biology, not the use of demographic methods or genomics (both of which have been around for more than a decade). I would drop the first paragraph and considerably expand the second. What has previous research on ecology/behavior/genetics found regarding the demographic effects of seasonal migration?

There are two important aspects to our study: (1) using paleodemographic methods to test hypotheses about adoption of a major life-history trait—an important biological question regardless of system, and so far (surprisingly) unaddressed; and (2) using this novel approach to study the effects of one such trait, seasonal migration. At these timescales, nothing exists on this subject, so there is really nothing to expand with. If there is relevant literature that we’ve missed, we’d be happy to add it.

What is the missing bit of information or angle the current study addresses (other than just doing it larger and fancier with genomics)?

The effects of major life-history traits on paleodemographics has not been addressed before, to our knowledge. The whole context is new, so we’re not doing something “larger and fancier” with genomics. We are doing something that has not been done before: testing hypotheses about the effects of a major life-history trait on population sizes in evolutionary time. We’re not sure how this can be made clearer. To us this seems like a very engaging biological question with wide applicability. We hope that this study is just the first of many to come, in a diversity of biological systems.

A figure showing the phylogenetic relationships of these taxa which are migratory would help the reader immensely. Although this is shown in Fig S3 I think it might be nice to have a map of the species and their ranges alongside a phylogeny as a main figure early on.

Thank you. This is a good suggestion. We can’t fit a phylogeny and all the distribution maps (Fig. S1) onto a page, but we can include a phylogeny as one of the main figures with nonmigrants highlighted. We’ve inserted this as a new Fig. 1.

If I understand correctly, the authors' arguments for why migratory species should show more growth hinge on large range size and geographic expansion. Yet they argue in the discussion that these forces are unlikely to be important (L226). I found the discussion on this confusing (e.g. L231 then says maybe it does matter). I think more clarity here would be helpful.

Our argument and predictions are based both on geographic and ecological expansion. This was clearly stated as our third prediction “3 early population growth would be higher as seasonal migration opens novel ecological and geographic space…” We have gone back through and reiterated the coupling of these two factors. The line mentioned concludes the first paragraph in the section ‘Geography and evolutionary ecology,’ which focuses on the difficulty of decoupling these in this system. As the paragraph relates, geography alone does not seem to be driving our results (we do not argue that it is unimportant).

I also would have liked more time in the discussion addressing why variation in *Ne* may be higher in migratory lineages.

In addition to re-clarifying this in the Introduction, we have touched back on this now at line 221: “We attribute the higher variation in *Ne* among migrants to be the result of the relative instability of northern biomes compared with tropical ones through glacial-interglacial cycles (e.g., Colinvaux et al., 2000; Pielou, 1991).”

Minor comments:L 62: Presumably PSMC is limited by the coalescent depth of the genelaogy, which may be younger or older than population "origins" depending on the history of colonization, lineage splitting, gene flow, etc.

We were careful to phrase these as “the lineages’ PSMC inception” (line 190), and responded to this issue in more detail above in response to R2’s public review.

L 338: I think a few more details on PSMC would be helpful. Was no maskfile used?

We did not use a maskfile, choosing instead to generate data of decent coverage and aligning reads to a single closely related relative.

Did the consensus fasta include all species?

No, we used a single reference high-quality fasta of *Catharus ustulatus* , as reported (lines 434-37). We have added that “Identical treatment of all lineages in these respects should provide a strong foundation for a comparative study like this among close relatives.”

L 361: Fair to assume the authors used a weighted average of *Ne* from the output, rather than just averaging the *Ne* values from each time segment?

No – we used all the values of *Ne* produced by PSMC output. The PSMC method uses nonoverlapping portions of the genome in its analyses (which we’ve added to make that clear), and portions in juxtaposition will often provide data for very different periods in the time segments. Further, time segments are uneven within and among taxa, so it is not clear how a uniform and comparable weighting scheme could be implemented. We consider a uniform approach to be of primary importance, including for future comparisons among studies.

L 383 "delta" typo

Thank you for catching this.

L 93: I'd be tempted to present the questions (how does seasonal migration affect population size trajectory, means, and variation) and rationale before presenting the hypotheses. I found myself reading the hypotheses and wondering "why?"

We’ve tried this change in the revision. It makes the hypotheses a little harder to pull out (they are no longer numbered in a short sequence), but it is shorter and solves this concern.

L 337 read depth is usually expressed as X (e.g. "23X") rather than bp.

Changed.